# Fluidic Oscillators, Feedback Channel Effect under Compressible Flow Conditions

**DOI:** 10.3390/s21175768

**Published:** 2021-08-27

**Authors:** Josep M. Bergadà, Masoud Baghaei, Bhanu Prakash, Fernando Mellibovsky

**Affiliations:** 1Fluid Mechanics Department, Universitat Politècnica de Catalunya, 08034 Barcelona, Spain; masoud.baghaei@upc.edu; 2Applus IDIADA Group, Santa Oliva, L’Albornar, 43710 Tarragona, Spain; bhanu.prakash@idiada.com; 3Department of Physics, Aerospace Engineering Division, Universitat Politècnica de Catalunya, 08034 Barcelona, Spain; fernando.mellibovsky@upc.edu

**Keywords:** fluidic oscillators, feedback channel performance, 3D-Computational Fluid Dynamics (CFD), Direct Numerical Simulation (DNS), compressible flow, boundary layer, flow control

## Abstract

Fluidic oscillators are often used to modify the forces fluid generates on any given bluff body; they can also be used as flow, pressure or acoustic sensors, with each application requiring a particular oscillator configuration. Regarding the fluidic oscillators’ main performance, a problem which is not yet clarified is the understanding of the feedback channel effect on the oscillator outlet mass flow frequency and amplitude, especially under compressible flow conditions. In order to bring light to this point, a set of three-dimensional Direct Numerical Simulations under compressible flow conditions are introduced in the present paper; four different feedback channel lengths and two inlet Reynolds numbers Re = 12,410 and Re = 18,617 are considered. From the results obtained, it is observed that as the inlet velocity increases, the fluidic oscillator outlet mass flow frequency and amplitude increase. An increase of the feedback channel length decreases the outlet mass flow oscillating frequency. At large feedback channel lengths, the former main oscillation tends to disappear, the jet inside the mixing chamber simply fluctuates at high frequencies. Once the Feedback Channel (FC) length exceeds a certain threshold, the oscillation stops. Under all conditions studied, pressure waves are observed to be traveling along the feedback channels, their origin and interaction with the jet entering the mixing chamber are thoroughly evaluated. The paper proves that jet oscillations are pressure-driven.

## 1. Introduction

Fluidic oscillators (FO) have a wide range of applications. They can be used to enhance mixing [1,2], as sensors to measure fluid flow [3,4,5], as fluidic sensors to measure micro/nanoscale transport properties [6], as heat transfer enhancers [7,8], and can serve as well as acoustic biosensors [9,10]. Their use to modify the boundary layer and therefore modify the forces acting on bluff bodies are among their most common applications [11,12,13]. Fluidic oscillators must be designed for each given application; this is why it is essential to understand the oscillator performance under different dimension modifications. Many of the studies on fluidic oscillators focused on two main very similar, canonical geometries, what Ostermann et al. [14] called the angled and the curved oscillator geometries. Some very recent studies using the angled geometry are the ones in [14,15,16,17,18,19,20,21,22,23,24,25]; the curved geometry was mainly studied by [1,14,26,27,28,29,30,31,32,33,34,35]. Ostermann et al. [14], compared both geometries, concluding that the curved one was energetically more efficient.

One of the first analyses of the internal flow in an angled fluidic oscillator was undertaken by Bobusch et al. [18]. Particle Image Velocimetry (PIV) experiments were performed using water as working fluid. The results provided detailed insight into the oscillation mechanism and also of the interaction between the mixing chamber fluid and the feedback channel flow.

Employing the same fluidic oscillator configuration previously analyzed in [18], Gartlein et al. [19] carefully evaluated the internal fluid structures as well as the outlet jet oscillation parameters using high speed PIV; they also used time-resolved pressure measurements. The Reynolds numbers studied ranged between 10,000 and 50,000, air was employed as working fluid. They observed a linear dependency between the oscillation frequencies and the input Reynolds number. Several fluid properties such as the deflection angle, jet width and jet velocity were examined. It was found that these properties remained rather constant for a certain range of Reynolds numbers, and suffered a strong change once a certain Reynolds number was overcome.

Gokoglu et al. [15] analyzed by using two-dimensional Computational Fluid Dynamics (2D-CFD), a very similar fluidic actuator configuration to the one evaluated by Bobusch et al. [18] and Gartlein et al. [19], but under supersonic flow conditions. Two fluids, helium and air, were considered as compressible and the k-omega SST turbulence model was employed for all the cases evaluated. The Mach numbers evaluated ranged between 0 and 2.5. No buffer zone was considered in the simulations. When evaluating the relation Strouhal versus Reynolds numbers and regardless of the flow conditions, subsonic or supersonic, a linear relation was observed. Under subsonic conditions, the Strouhal dependency on Reynolds number was much more relevant for helium than for air; such dependency was less relevant at supersonic conditions. It was observed that the time required for the oscillations to start sharply decreased as the Mach number increased, this time remaining almost constant at supersonic flow conditions.

The same configuration previously employed in [18], although now using a single exit, was numerically evaluated in 3D at Reynolds 30,000 and using the k-omega SST turbulent model by Pandley and Kim [22]. Two geometry parameters, the mixing chamber inlet and outlet widths were modified. They observed a significant effect of the flow structure and the feedback channel flow rate when modifying the inlet width, while negligible effects were observed when modifying the outlet width.

Woszidlo et al. [20] studied the same configuration previously evaluated by Gartlein et al. [19]. Both configurations resemble the one studied by Bobbush et al. [18], the main differences resided in the outlet shape. In Woszidlo et al. [20] and Gartlein et al. [19], just a single outlet was considered. In this new paper, Woszidlo et al. [20] focused their attention on analyzing the flow phenomena inside the mixing chamber (MC) and the feedback channels (FC). They also highlighted that the increase of the MC inlet width was tending to increase the outlet frequency and rounding the feedback channels diminished the generation of the separation bubbles on these channels’ corners.

The fluidic oscillator outlet frequency and amplitude, whenever the feedback channel and the mixing chamber lengths were modified, was studied using a 2D numerical model by Seo et al. [23]; the fluid was considered as incompressible. An increase of the feedback channel length generated no modifications on the outlet frequency, and the same observation was previously obtained by [36]. In both cases, the flow was considered as incompressible, which is likely to be the reason why such outcome was obtained. Slupski and Kara [31] studied using 2D-URANS with the software Fluent a range of feedback channel (FC) geometry parameters, and the sweeping jet actuator configuration was the same as the one analyzed by Aram et al. [32]. The effects of varying the feedback channel height and width for different mass flow rates were studied. All the simulations were performed for a fully turbulent compressible flow, using k-omega SST turbulence model. It was found that oscillation frequencies increased with increasing feedback channel height up to a certain point and then remained unaffected; however, frequencies decreased by further increasing the feedback channel width.

Via using stereoscopic PIV, the velocity field properties emitted by an FO rounded configuration were studied by Ostermann et al. [1] at Reynolds < 50,000. Among the conclusions of their study, the necessity of performing compressible flow simulations to properly understand the flow physics inside the oscillator was observed.

An experimental and numerical study of a fluidic oscillator which could generate a wide range of frequencies (50–300 Hz), was studied by Wang et al. [37]. Their study focused on the oscillation frequency response for different lengths of the feedback channels. 2D compressible simulations were performed using sonicFoam with k-epsilon as turbulence model. An inverse linear relation between frequency and the length of feedback loops was observed, whereby frequency increased by decreasing the feedback channel length. In Hirsch and Gharib [38], the dynamics of a sweeping jet actuator were analyzed via Schlieren visualizations. Subsonic Mach numbers and the transition to sonic conditions were evaluated. They observed that the oscillations started from small asymmetries caused by small differences in geometry.

The present paper is presenting a 3D-DNS numerical evaluation of the same fluidic oscillator configuration employed in [18]. The effect on the oscillator dynamic characteristics of four different feedback channel lengths and two inlet velocities, will be be studied. The flow is considered as compressible and subsonic, air being the working fluid. The remaining of the paper is designed as follows. The introduction of the non-dimensional groups is followed by the mathematical background and the mesh independence study. After that, the results section and the paper’s conclusions are the two final sections.

## 2. Dimensional and Non-Dimensional Parameters

Despite the fact that some of the graphs introduced in the present paper are presented in dimensional form, it is necessary to define the different parameters used for non-dimensionalization as well as the non-dimensional numbers used in the present manuscript. The Reynolds number definition used to characterize the main flow was $R_{e}=\frac{\rho VD_{h}}{\mu }$, ρ and μ are the density and dynamic viscosity of the fluid, respectively, $D_{h}$ is the hydraulic diameter and *V* is the spatial averaged fluid velocity; in the present paper all these parameters were defined at the FO power nozzle, see Figure 1. The definition employed for the Mach number was $M=\frac{V}{C}$, where for an ideal gas $C=\sqrt{\gamma RT}$, *T* being the fluid temperature, *R* the gas constant and γ the adiabatic index. Despite the fact that the Mach number and the Reynolds number are defined at the power nozzle, the maximum fluid velocity was found under all conditions evaluated, at the MC outlet width, which is why the maximum Mach number is defined based on the fluid conditions in this particular section. The non-dimensional wall coordinate $y^{+}$ is defined as $y^{+}=\frac{yu_{\tau }}{\nu }$, $u_{\tau }=\sqrt{\frac{\tau_{w}}{\rho }}$, where *y* is the distance between the first mesh cell and the wall, $u_{\tau }$ is the friction velocity, ν stands for the kinematic viscosity and the term $\tau_{w}$ characterizes the wall shear stresses. The non-dimensional frequency is defined as $F^{+}=\frac{fD_{h}}{V}$, *f* being the dimensional frequency. The advective time is given according to the following equation—$\tau =\frac{TV}{D_{h}}$ where *T* defines the dimensional time.

Another parameter particularly useful in the present manuscript is the momentum *M* associated to the fluid at the (FC’s) outlet, see Figure 1e. Notice that the momentum associated to the fluid in a given section characterizes the total force the fluid is capable of achieving. According to the momentum equation it can be stated:(1)$$M=\dot{m}V+PS={\dot{m}}^{2}/\left(S\rho \right)+PS$$
where $\dot{m},V,S$, and *P* are, respectively, the mass flow, the spatial averaged fluid velocity, the surface where the momentum needs to be evaluated and the instantaneous spatial averaged static pressure; ρ is the spatial averaged fluid density.

As the flow is considered as compressible (fluid density, velocity and pressure are spatial and temporal dependent), the instantaneous momentum equation to be applied at any given surface (FC’s outlet in the present paper) needs to be discretized and evaluated at each grid cell belonging to the corresponding surface.
(2)$$M=M_{massflow}+M_{pressure}=\sum_{i=1}^{i=n}\left(S_{i}\rho_{i}V_{i}^{2}\right)+\sum_{i=1}^{i=n}P_{i}S_{i}$$

The subindex *i* denotes any parameter associated to the generic mesh cell belonging to the FC outlet surface under evaluation. The term *n* defines the total number of cells corresponding to any of the two FC outlet surfaces.

Based on the previous equation, it can be seen that the momentum associated to the flow at any given section consists of two parts, the momentum due to the fluid mass flow and the one associated to the static pressure. In order to evaluate the mass flow momentum term, it is necessary to know the instantaneous fluid density and velocity associated to each mesh cell section through which the fluid flows. In the present paper, the instantaneous momentum term due to the mass flow was obtained simply by adding the elementary momentum terms of each mesh cell belonging to the chosen surface. The momentum pressure term was obtained when multiplying the instantaneous static pressure acting on each cell by the cell surface, and then adding the elemental momentum pressure terms corresponding to the surface under study.

The two instantaneous momentum terms were obtained at each of the two feedback channel outlets. The instantaneous net-momentum, characterizing the overall forces acting on the main jet lateral surfaces (the two FC outlets), considers the instantaneous pressure and mass flow momentum terms acting on both feedback channels’ outlets.

To be able to compare the relation between several parameters triggering the oscillations, some of the graphs presented were made non-dimensional. The non-dimensionalization was done by dividing each variable by the maximum value of the same variable measured at the same location and obtained from the baseline case oscillator at the smallest power nozzle velocity studied, 65 m/s. Two power nozzle velocities of 65 m/s and 97 m/s are reported in the present manuscript.

## 3. Mathematical Background

In the present manuscript, the three-dimensionalcompressible Navier–Stokes equations are employed as the governing equations. In Cartesian coordinates, such equations take the following form.
(3)$$\frac{\partial Q}{\partial t}+\frac{\partial E}{\partial x}+\frac{\partial F}{\partial y}+\frac{\partial G}{\partial z}=\frac{\partial E_{v}}{\partial x}+\frac{\partial F_{v}}{\partial y}+\frac{\partial G_{v}}{\partial z}$$

The term Q characterizes the conservative variables, E, F and G define the *x*, *y* and *z* components of an inviscid flux, respectively. The vectors $E_{v}$, $F_{v}$ and $G_{v}$ represent the *x*, *y* and *z* components of a viscous flux, respectively. The definition of all these vectors is presented next.
(4)$$\begin{array}{cc} \; & Q={\left(\rho \;\rho u\;\rho v\;\rho w\;e\right)}^{T},\\ \; & E={\left(\rho u\;\rho u^{2}+p\;\rho uv\;\rho uw\;\left(e+p\right)u\right)}^{T},\\ \; & F={\left(\rho v\;\rho vu\;\rho v^{2}+p\;\rho vw\;\left(e+p\right)v\right)}^{T},\\ \; & G={\left(\rho w\;\rho wu\;\rho wv\;\rho w^{2}+p\;\left(e+p\right)w\right)}^{T}, \end{array}$$
(5)$$\begin{array}{cc} \; & E_{v}={\left(0\;\tau_{xx}\;\tau_{xy}\;\tau_{xz}\;\beta_{x}\right)}^{T},\\ \; & F_{v}={\left(0\;\tau_{yx}\;\tau_{yy}\;\tau_{yz}\;\beta_{y}\right)}^{T},\\ \; & G_{v}={\left(0\;\tau_{zx}\;\tau_{zy}\;\tau_{zz}\;\beta_{z}\right)}^{T}, \end{array}$$
(6)$$\begin{array}{cc} \; & \beta_{x}={\left(\tau_{xx}u+\tau_{xy}v+\tau_{xz}w-q_{x}\right)}^{T},\\ \; & \beta_{y}={\left(\tau_{yx}u+\tau_{yy}v+\tau_{yz}w-q_{y}\right)}^{T},\\ \; & \beta_{z}={\left(\tau_{zx}u+\tau_{zy}v+\tau_{zz}w-q_{z}\right)}^{T}, \end{array}$$

The variable ρ stands for the fluid density, which for the present simulations is considered spatial and temporally dependent; *u*, *v* and *w* represent the fluid velocity components in the *x*, *y* and *z* directions, respectively. $\tau_{ij}$ is a generic component of the viscous stress tensor, $q_{x}$, $q_{y}$ and $q_{z}$ are the heat fluxes in the *x*, *y* and *z* directions, respectively.

The stress tensor τ is a linear function of the strain rate tensor or equivalently the velocity gradient. The equation used to determine each term of the stress tensor is:(7)$$\tau_{ij}=\mu \left(\frac{\partial u_{i}}{\partial x_{j}}+\frac{\partial u_{j}}{\partial x_{i}}\right)+\delta_{ij}\lambda \left(\frac{\partial u_{k}}{\partial x_{k}}\right)$$
where $\lambda =-\frac{2}{3}\mu$ and $\delta_{ij}$ is the Kronecker delta. The total energy per unit volume *e* is defined in terms of the following equation of state for ideal gas.
(8)$$e=\frac{p}{\gamma -1}+\frac{1}{2}\rho \left(u^{2}+v^{2}+w^{2}\right)$$
where *p* and γ are the pressure and the specific heat ratio, respectively. For the present study, air was used as working fluid and it was considered as a Newtonian and ideal gas. The oscillator walls were considered as adiabatic, the specific heat ratio was 1.4. Dynamic viscosity was considered as constant and Prandtl number was set to 0.7.

Direct Numerical Simulation (DNS) of turbulent flow takes this set of equations as a starting point and develops a transient solution on a sufficiently fine spatial mesh with sufficiently small time steps to resolve even the smallest turbulent eddies and the fastest fluctuations. The feasibility of these requirements is evaluated in the next section.

## 4. Mesh Assessment

Figure 1a–c introduce the 3D medium mesh employed for the baseline FO. In Figure 1a, the main parts of the FO are presented along with the entire mesh, Figure 1b shows the mesh inside the MC and the FCs, the vertical line depicted in Figure 1b shows the exact location of the mesh section presented in Figure 1c. The definition of the main geometry parameters is presented in Figure 1e.

The different boundary conditions employed in all simulations were Dirichlet boundary condition (BC) for velocity and temperature at the FO inlet, Neumann BC for pressure. At the FO outlet, Neumann BC for temperature and velocity as well as Dirichlet BC for pressure were used. At the walls, Neumann BC for pressure and temperature as well as Dirichlet BC for velocity were chosen. The time step used for all simulations was of $2.25\times 10^{-8}$ s. Second-order discretization was used for all parameters except for time and convection where first-order discretization was employed. Two fluid velocities of 65 m/s and 97 m/s defined at the FO power nozzle were considered. The respective Reynolds and corresponding Mach number at the same section were Re = 12,410, Re = 18,617, *M* = 0.188 and *M* = 0.282. Based on these Mach numbers, the fluid could nearly be considered as incompressible, yet as it will be presented in results section, the maximum Mach number observed inside the mixing chamber leave no doubt regarding the need to consider the fluid as compressible.

To to the authors’ knowledge, on the present fluidic oscillator configuration, no previous 3D-DNS simulations considering the fluid as compressible have been undertaken. In the present paper, and knowing that for the operating conditions chosen no comparisons with previous experimental or numerical results are possible, initially a mesh independence study using the baseline fluidic oscillator configuration for a velocity of 97 m/s measured at the fluidic oscillator power nozzle, see Figure 1e, was performed. Three different mesh densities were evaluated, their respective number of cells were 5,933,900, 19,600,672 and 23,882,040, being their respective maximum Y+, obtained after the simulations of 23.86, 3.16 and 2.6. For each mesh, the FO upper outlet mass flow oscillating frequency was measured and is presented in Figure 1d, where it is observed that the FO oscillating frequency has a relatively small dependency on the mesh density chosen. When comparing the results obtained using the medium and the finest mesh, the frequency varied 0.96%, the variation increased to 5% when the coarsest and the finest mesh results were compared. The authors decided that the medium mesh, having 19,600,672 cells, was accurate enough for the present purposes and it was used for the rest of the simulations.

Figure 2 introduces the different feedback channel lengths evaluated in the present study. Defining L as the feedback channels length for the baseline case, the other three FCs represented in Figure 2b–d, have lengths of 2L, 3L and 9L, respectively, the rest of the FO dimensions were kept constant for all cases. The number of cells employed for the FO with different feedback channels lengths, represented in Figure 2b–d, were respectively of 21,736,832, 22,394,112 and 32,417,632. The mesh in the FO main body remained the same in all cases, just the FC’s mesh was modified to accommodate the different FC lengths. The main dimensions of the FO are introduced in Figure 3, their respective values are: Ya=0.0237 m, Yb=0.0216 m, Xa=0.0198 m, Xb=0.0389 m, Da=0.00255 m and Db=0.0034 m. The depth of the FO, which can be observed in Figure 1c, is constant at all points, its value is of 0.00325 m.

As compressible three-dimensional Direct Numerical Simulations (3D-DNS) are employed in all the cases studied, and in order to prove the simulations can be considered as DNS, Figure 3a representing the energy decay inside the FO was generated. Two probes (A, B), located respectively at $A_{probe}\Rightarrow (x=0.06642\;m,y=0.001693\;m,$z=0.00195m) and $B_{probe}\Rightarrow \left(x=0.06642\;m,y=0.010157\;m,z=0.00195\;m\right)$ were employed as representative to evaluate the fluid turbulent kinetic energy. The location of these two probes is presented in Figure 3b. The red lines shown in the same figure represent the plane probe locations at the MC inclined walls, the FC outlets and the FO upper outlet. From the energy spectrum, the inertial subrange zone, having a pendent of −5/3, is clearly observed; the dissipation range which appears at the end of the inertial subrange can also be seen. Notice that when the dissipation range is clearly observed, it means the mesh cells have a characteristic length of the order of the Kolmogorov length scales, and so the simulation is capable of evaluating (to a certain extent) the energy dissipation of the smallest vortical structures. This fact is characteristic of DNS simulations. Another variable, used as a rule of thumb to identify DNS simulations, is the relation between the energy level associated to the smallest flow structures and the largest ones. When about six orders of magnitude are separating these two different energy levels, simulations can be considered as DNS. Notice that this is exactly what the energy spectrum is showing in Figure 3. The conclusion is that the simulations about to be introduced can be considered as DNS.

A final point to consider is the viability of employing a buffer zone. In previous studies [39], a buffer zone was employed to increase the accuracy of the simulations, yet it was observed that the buffer zone affected in a small percentage the outlet mass flow and its associated frequency. In the present manuscript and due to the fact that simulations are 3D-DNS, it was decided not to use a buffer zone, as the number of cells would have increased enormously and so would have the computational time. The computational time to complete one simulation was about 47 days using 96 cores (two nodes of 48 cores) in a supercomputer based on two 24 cores Intel Xeon Platinum processor. A time step of $2.25\times 10^{-8}$ s/iteration was employed in all simulations. To create the different meshes, the package (GMSH) was used, the finite volumes-based software (OpenFoam) was employed to perform the simulations.

## 5. Results

### 5.1. Baseline Case at Two Different Power Nozzle Velocities

In the present subsection, the main flow parameters associated to the FO working under compressible flow conditions and at the two Mach numbers evaluated are introduced. The initial analysis is focused on the baseline case. The main vortical structures inside the FO mixing and external chambers, given for a period of oscillation ξ divided in six equally spaced time steps, are presented based on the Q criterion in Figure 4 and Figure 5 for the two power nozzle velocities evaluated, 65 m/s and 97 m/s, respectively. Both figures show how the positive and negative vortical structures are alternatively appearing inside the mixing chamber and being shed downstream. The Q criterion evaluates the relative dominance of the rotational over the stretching component in the deformation of a fluid element, mathematically defined as $Q=0.5({vorticity}^{2}-{strain\;rate}^{2})$. Notice that the vorticity is defined as angular momentum in radians per second. Particularly large positive three-dimensional vortical structures inside the MC are clearly seen at time periods ξ=0, ξ=2/3 and ξ=5/6, large negative vortical structures are observed at time periods ξ=1/6 and ξ=1/3. Similar structures on the same FO configuration but under incompressible flow conditions were recently presented in [24,39]. The dimension of the vortical structures is very much the same for the two velocities studied, but the maximum vorticity associated to the main vortical structures increases about 66% when the power nozzle velocity is increased from 65 m/s to 97 m/s. Notice that the vortical structures having associated the maximum and minimum vorticity values, appear alternatively inside the mixing chamber, just after the MC inlet width, in the locations where the minimum pressure is observed, see Figure 6 and Figure 7. The main vortical structures break and become more random as they are being shed downstream. Inside the feedback channels, weak vortical structures appear alternatively at the FC’s inlet and also at the opposite FC left hand side corner, see Figure 4b,e and Figure 5b,e, indicating the locations where the flow is generating eddies. As already observed by [20], the FC’s eddies could be minimized by rounding the FC’s corners. Rather weak vortical structures appear at the external chamber (EC), indicating that although large vortices are generated in this chamber, see Figure 6a and Figure 7a, their intensity is much weaker than the vortical structures generated inside the MC.

Figure 6 and Figure 7 show the instantaneous velocity, pressure, density and temperature fields; in the FO mixing and external chambers, the power nozzle velocities are 65 m/s and 97 m/s, respectively. A three-dimensional Q criterion plot generated at the same instant is as well presented in each figure. The time instant shown in Figure 6 and Figure 7 corresponds to the time period ξ=0 presented in Figure 4 and Figure 5, respectively. It is interesting to realize that the fluid expands inside the mixing chamber, the pressure decreases about 10,000 Pa in Figure 5 and about 13,000 Pa in Figure 6, versus the respective one at the power nozzle. The maximum velocity and Mach number at the MC outlet reach values 35% and 44% higher than the ones at the power nozzle, respectively. The fluid suffers a second expansion when it reaches the external chamber, the pressure being rather homogeneous across it. In both Figure 6a and Figure 7a, a large vortical structure of nearly the same size is observed at the external chamber upper side. When comparing these two figures it can be clearly stated that the vortical structure shown in Figure 7 has a higher turning speed associated than the one appearing in Figure 6. In fact, in Figure 7, the pressure at the vortex central core is about 3000 Pa lower than the one observed at the surrounding area, this pressure drop is about 1000 Pa in Figure 6.

The feedback channels are alternatively pressurized, but due to the fluid compressibility, it can be seen that just parts of the feedback channels are pressurized; in other words, pressure waves can be seen moving along the FCs. The videos presented in Appendix A are designed to help in clarifying this point. Pressure waves traveling along FCs were very recently reported by [40]. At the instant presented in Figure 6, the lower FC outlet is slightly pressurized while the upper one is not, therefore suggesting that the jet inside the mixing chamber is being pushed upwards. Figure 7 shows the opposite, the upper FC outlet is slightly pressurized while the lower one is not, indicating the main jet entering the MC is at this instant being pushed downwards. At this point it needs to be clarified that although Figure 6 and Figure 7 are defined at time ξ=0, this instantaneous initial time represents different physical conditions for each figure.

Pressure waves originate at the MC converging walls and are due to the stagnation pressure points appearing alternatively at these surfaces. When carefully looking at Figure 6b–d and Figure 7b–d, it is seen that pressure waves also originate at the FC’s outlets, see the small red point at the lower FC outlet internal vertical wall, Figure 6b. Notice that from this point pressure waves are being generated and move from the FC outlet towards the FC inlet. Figure 6b clearly shows the existence of two pressure waves, one is moving along the upper FC, from FC inlet to the outlet and the other is moving along the lower FC from the FC outlet to the FC inlet. The pressure wave generated at the FC outlet has associated a smaller intensity than the other one. Although the maximum value of the stagnation pressure was found to be at the FC outlet, the time during which the stagnation pressure point is observed is much smaller than the time stagnation pressure is seen to last at the MC converging walls. This explains the different intensities of the pressure waves generated on each FC end. As will be clarified in the remaining part of the paper, whenever pressure waves originate at each side of the FCs, the resulting net momentum acting over the main jet as well as the stagnation pressure dynamics at the MC converging walls have associated particularly high fluctuations.

Pressure waves generated at the MC converging walls do not appear to be moving downstream towards the external chamber; it seems that the main jet flowing across the mixing chamber outlet width blocks them from displacing downstream. The same happens to the pressure waves generated at the FC outlets’ internal vertical walls. The waves propagate along the FC but not towards the MC, the main jet entering the MC appears to be acting as a barrier to the pressure wave propagation. On the other hand, some high-frequency waves are likely to dissipate into heat. The fluid expansion suffered at the EC tends to help in dissipating the pressure waves effect in this chamber. The Q criterion plots presented in Figure 6e and Figure 7e show the three-dimensional structures already presented in two dimensions in Figure 4a and Figure 5a, respectively. Notice that the main three-dimensional structures, alternatively appearing at the MC inlet, maintain their form along the spanwise direction. Four videos introducing the velocity field and the pressure contours characterizing the two power nozzle velocities studied and for the baseline case are presented in the Appendix A.

When studding the same FO configuration under incompressible flow conditions and at Reynolds numbers up to 32,000, refs. [39,41] established that flow oscillations were driven by the alternating pressure variation at the FC’s outlets. Based on the results presented in this sub section, the required force responsible for driving the jet oscillations inside the MC appears to be generated by the pressure difference at the FC’s outlets. This hypothesis shall be proved in the remaining part of the present manuscript.

In order to determine the origin of the forces responsible of the jet oscillations inside the MC, Figure 8 was created. It presents for the two velocities studied and in non-dimensional form, the dynamics of the maximum stagnation pressure measured at the MC lower converging wall, the FO upper outlet mass flow, the lower FC mass flow and the net momentum applied to the jet entering the MC. Figure 8a,b characterize the baseline case, and Figure 8c,d introduce the same information for the maximum feedback channel length studied, 9L. The first thing to realize is that, for the baseline case and for both power nozzle velocities, all these parameters follow the same trend. On the other hand, for the longest FC length studied, the main jet oscillations are deeply affected by random fluctuations, and under these conditions the main jet inside the MC is mostly fluctuating although performing as well a low amplitude oscillation. In Figure 8a,b, the stagnation pressure oscillations measured at the MC lower converging wall appear at the same frequency as the rest of the parameters, indicating that there must be a correlation between them. The amplitude of these parameters also appears to be correlated. The oscillating frequencies of all these parameters when the power nozzle velocities are 65 m/s and 97 m/s are respectively of 275.4 Hz and 410 Hz.

When the power nozzle inlet velocity is of 65 m/s, Figure 8a, the stagnation pressure oscillations are particularly scattered, the FC mass flow and in particular the net momentum acting on the jet entering the MC show as well as very scattered curves, which appear to be affected by very high frequency fluctuations. For this particular inlet velocity, as shown in Figure 6b, a stagnation pressure point is observed at the FC lower outlet internal vertical wall. From this point, pressure waves are being generated and move from the FC outlet towards the FC inlet. These pressure waves interact with the ones generated at the MC converging walls and create the high-frequency fluctuations observed in the pressure, net momentum and FC mass flow curves shown in Figure 8a. All these pressure waves bounce on the different FC walls which help in generating high-frequency fluctuations.

As the power nozzle velocity increases to 97 m/s, the stagnation pressure point appearing at the FC outlet internal vertical wall reaches a smaller value than in the previous case, compare Figure 7b,c with Figure 6b,c. Fewer fluid particles from the incoming jet are impinging in this vertical wall, and as a result the pressure waves generated at this particular point are weaker than in the previous velocity studied. The result is that a much smoother stagnation pressure oscillation is observed at the MC converging walls and at the FC mass flow, the net-momentum measured at the FC outlet surfaces is also affected by less fluctuating perturbations, see Figure 8b. The fluctuations associated to all these curves show a much smaller frequency than in the previous case. Yet, the net momentum acting on the MC incoming jet is still showing large amplitude fluctuations, which is due to the pressure waves periodically generated at the MC converging walls and bouncing on the FC’s walls, alternatively pressurizing and depressurizing the FC’s outlets. For a power nozzle velocity of 65 m/s, the stagnation pressure peak-to-peak oscillations amplitude are about 3.5% of the maximum stagnation pressure at this power nozzle velocity. The non-dimensional peak-to-peak stagnation pressure oscillations amplitude increases to about 5% when the power nozzle velocity reaches 97 m/s. In reality, both pressure waves show a main oscillating peak and a smaller one, this second one coincides with the instant the main jet impinges on the opposite MC converging wall, therefore pressurizing the opposite FC. The increase of the peak-to-peak stagnation pressure amplitude at the MC converging walls is triggering the amplitude increase of the FO outlet mass flow, the FC mass flow and the net momentum evaluated at the FC outlets. Notice that all these parameters show a much larger peak-to-peak amplitude in Figure 8b when compared to the ones in Figure 8a. The net-momentum peak-to-peak amplitude at 97 m/s almost doubles the value observed at 65 m/s. From these two figures it can also be observed that for any of the two velocities studied, the FO outlet mass flow, the FC mass flow and the net momentum have, during approximately 1/4 of the oscillating cycle, negative values. This clearly indicates there is reverse flow at the FO outlet and inside the FCs. Reverse flow at the lower FC outlet can be clearly seen in Figure 6a. The increase of power nozzle velocity does not seem to affect much the minimum negative values of the FC mass flow and the FO mass flow. Even the time oscillating parameters remain negative, 1/4 of the oscillating cycle appears to be unaffected by the power nozzle inlet velocity.

### 5.2. The Effect of the Feedback Channel Length

From the comparison of Figure 8a,b, baseline case, with Figure 8c,d, feedback channel length 9L, it is observed that regardless of the inlet velocity employed, the FO outlet flow drastically reduces its peak-to-peak amplitude when the longest feedback channel length is used. Although the different graphs are not presented in this paper, it was observed that the stagnation pressure oscillation amplitude at the MC converging walls decreased as the feedback channel length increased. The second major observation from Figure 8 is that unlike in the baseline case, for the FC length 9L, there is not an apparent link between the stagnation pressure at the MC convergent walls and the rest of the parameters. However, when closely looking at the different curves presented in Figure 8c, it can be observed the different curves tend to follow, although with a phase lag, the stagnation pressure oscillations appearing at the MC converging wall. Again this appears to indicate that the origin of the jet oscillations and even the fluctuations is the stagnation pressure variations at the MC converging walls. For a FC length of 9L and for both velocities studied, the jet inside the MC is mostly fluctuating at high frequencies, the fluctuation amplitude is of the same order of the oscillation amplitude. The fluctuation frequency increases with an increase of the inlet velocity; in fact, for the highest velocity and FC length studied, Figure 8d, random fluctuations dominate the flow, high Reynolds numbers and long FC lengths seem to enhance flow randomness.

To be able to further understand the effect of the FC length on the FO outlet mass flow frequency and amplitude, Figure 9 was generated. Regardless of the inlet velocity employed, as the FC length increases, the FO outlet mass flow oscillating frequency and its peak-to-peak amplitude tend to decrease. At 65 m/s, the equations characterizing the FO outlet mass flow amplitude and frequency decrease, normalized by its maximum respective value, and as a function of the FC length increase are respectively: $A_{normalized}=0.00601399\times {\left(Li\right)}^{3}-0.0886317106\times {\left(Li\right)}^{2}+0.2381284122\times \left(Li\right)+0.8444893085$; $f_{normalized}=0.9620906099\times {\left(Li\right)}^{-0.0326731141}$, where $L_{i}$ take the values of the different FC lengths, 1, 2, 3 and 9. The evolution of the rest of the parameters, net-momentum and stagnation pressure at the MC converging walls, follow exactly the same trend. Although they used a different FO configuration, the oscillating frequency variation is in full agreement with the observations made by [37].

In fact, and regardless of the velocity studied, the curves showing the FO outlet mass flow become scattered as FC length increases. It appears as if instead of clean oscillations, the main jet undergoes some vibrational movement, although still maintaining a low amplitude oscillatory displacement, see Figure 8c,d and Figure 9b. For a power nozzle velocity of 65 m/s, Figure 9a, the jet fluctuations (vibrational movement) can be clearly seen when the FC length is 3L; they are almost as large as the peak-to-peak oscillation amplitude when the FC length is 9L. Under these conditions, the main jet at the FO outlet undergoes a high-frequency fluctuating movement while performing a small oscillation cycle. For a FC length of 9L, the frequencies associated to the respective fluctuating and oscillating jet movements measured at the FO outlet mass flow are 1037.4 Hz and 251.6 Hz. When the power nozzle velocity is of 97 m/s, the fluctuating wave can be clearly seen for a FC length of 2L, see Figure 9b. Whenever the FC length is 3L, the jet fluctuations have a peak-to-peak amplitude of about 25% of the oscillation one. For this particular case, the frequencies associated to the fluctuating and oscillating movements are 2401.8 Hz and 179.3 Hz, respectively. For the longest FC length L9, the peak-to-peak amplitude associated to the jet fluctuations are smaller than for the FC length L3; in reality the fluctuations as well as the oscillations have become quite random, a set of different frequencies appear, the frequencies associated to the major flow fluctuations and to the jet main oscillating displacement are respectively of 2708.8 Hz and 132.1 Hz. In any case, it can be concluded that the randomness associated to the oscillations increases with the FC length increase, and/or with the FO inlet velocity increase. Based on these results, it can be estimated that a further increase of the FC length would make the oscillations stop, as it is about to happen in Figure 9b for the maximum FC length L9. The simulations demonstrate that reverse flow has to be expected at the FO outlet, the reverse flow at the FO outlet tends to disappear as the FC length increases. Figure 9a,b demonstrate there is no reverse flow at the FO outlet for a FC length of 9L.

Figure 10 presents for the two power nozzle velocities and the four FC lengths the evolution of the oscillating and fluctuating frequencies, as well as their respective amplitudes. The information is obtained based on the FO outlet mass flow. At 65 m/s, the oscillation amplitude decreases by about 80% when comparing the results for the longest FC length 9L with the baseline case. For the same conditions, the oscillation frequency decreases by 9%. For a FC length of 9L, a clear fluctuating frequency of 1037.4 Hz, which is superposed to the oscillating wave, can be observed. Whenever the power nozzle velocity is of 97 m/s, Figure 10b, and when comparing the results of the baseline case with the ones for the longest FC length, the oscillation amplitude decreases by about 72% and the frequency by 67%. At high speeds, the compressibility effect along with the chaotic stage make the oscillating frequency highly dependent on the FC length, generating a drastic decrease of the oscillating frequency and the appearance of several fluctuating frequencies. Especially at 97 m/s, the outlet mass flow fluid oscillations become chaotic as the FC length increases, oscillations are rather chaotic for a FC length of 9L. As observed in Figure 10, fluctuating frequencies are one or several orders of magnitude higher than the oscillation ones.

Once the main FO outlet flow characteristics are evaluated, it is interesting to analyze the fluid evolution inside the MC and the feedback channels. For the two power nozzle velocities and the four FC lengths, the overall forces acting on the main jet at the MC inlet, expressed as the net-momentum acting onto the FC’s outlets are presented in Figure 11. For the two power nozzle velocities studied, the net momentum oscillation amplitude and frequency associated decreases with the FC length increase. The net-momentum oscillation becomes highly random as the FC length increases, the inlet velocity increase enhances this effect. At high FC lengths, large amplitude pressure fluctuations are mostly observed on the MC converging walls, see Figure 8c,d, and these fluctuations take control of the main flow and of the net momentum acting on the jet, as observed in Figure 11b. For the baseline case and for the smaller velocity studied, 65 m/s, Figure 11a, the net-momentum oscillating displacement appears to be particularly noisy, the frequency associated to the fluctuations (noise) is of 9988.6 Hz, about 35 times the oscillating frequency, which is of 275.4 Hz. This very high frequency appears to be associated to the pressure waves generated at the MC inclined walls and to the bouncing of the pressure waves on the FC walls. In the previous section it was already explained that under these conditions, a particularly high stagnation pressure point appears at the FC’s outlet internal vertical wall, see Figure 6. Pressure waves are simultaneously generated at this point and at the MC converging walls, the interaction between these pressure waves, which displaces in opposite directions, is the origin of these particular high-frequency fluctuations observed in Figure 11a for a FC length L1. Figure 11b for FC lengths L3 and L9, show that the main net-momentum oscillation is almost gone and the main jet is affected by very high frequency fluctuations, of which the peak-to-peak amplitude is larger than the oscillating one. Notice that for these particular cases, the FO outlet mass flow, Figure 9b, presents high-frequency fluctuations superposed on a highly random oscillation. Notice as well that the net-momentum pattern has close similarities with the outlet mass flow pattern and the mixing chamber pressure one on the converging walls. To highlight these similarities, Figure 12 and Figure 13 were generated.

Figure 12 introduces the Fast Fourier Transformation (FFT) of the net-momentum signals determined at the FC’s outlets. The results for both power nozzle velocities and the four FC’s lengths are presented. From the comparison of Figure 10 and Figure 12, it is seen that the main oscillating frequencies presented in Figure 10 are also observed in Figure 12. This means there must be a correlation between the output mass flow and the net momentum acting on the jet entering the MC (FC’s outlet). At the FC’s outlets, several high-frequency peaks not appearing at the FO outlet are observed. These very high fluctuation frequencies are likely to be associated to the pressure waves traveling along the FC’s, but they do not seam to be transferred to the FO outlet. It looks as though the main jet flowing thought the MC outlet width acts as a barrier for most of the fluctuating waves existing inside the MC. In fact, part of the energy associated to these very high frequency fluctuations may dissipate into heat inside the MC and FCs.

In order to prove that the origin of FO oscillations is the stagnation pressure temporal evolution at the MC converging walls, the FFT of the stagnation pressure signal measured at the MC lower converging wall is presented in Figure 13. The four FC lengths and the two inlet velocities are considered in this figure. From the comparison of Figure 10, Figure 12 and Figure 13, it can be concluded that the main oscillating frequencies appear in the three figures, proving that there must be a correlation between the stagnation pressure, net momentum acting onto the jet entering the MC and the FO outlet mass flow. High fluctuating frequencies are observed in Figure 12 and Figure 13 but hardly in Figure 10, suggesting that the jet fluctuations travel along the FCs but barely downstream.

Another important point to clarify is which of the two terms characterizing the net momentum acting onto the jet is the dominant one. In a previous numerical work done by Baghaei and Bergada [39,41], using the same oscillator configuration but under incompressible flow conditions, water was used as working fluid, it was stated that the net-momentum pressure term, see Equations (1) and (2), was the dominating one and therefore was responsible for the main jet oscillations. Under compressible flow conditions, Figure 14 compares for the baseline case and the maximum FC length L9, the net-momentum mass flow term $M_{massflow}=\sum_{i=1}^{i=n}\left(S_{i}\rho_{i}V_{i}^{2}\right)$ with the net-momentum pressure term $M_{pressure}=\sum_{i=1}^{i=n}P_{i}S_{i}$ measured at the FC outlets. This figure clearly shows that regardless of the feedback channel length and the power nozzle velocity, the net momentum due to the pressure term is over an order of magnitude higher than the one generated by the mass flow term. The conclusion is that the forces driving the oscillations in this particular FO configuration are due to the pressure difference acting at the FC’s outlets. Modifying the FC length, the inlet velocity or evaluating the fluid under compressible or incompressible conditions does not change this statement.

In order to visualize the flow topology for a FC length of 9L, instantaneous velocity, pressure, density and temperature fields and for the two power nozzle velocities of 65 m/s and 97 m/s are presented in Figure 15 and Figure 16, respectively. Four videos showing the velocity and pressure fields for each power nozzle velocity and for a feedback channel length of 9L are also provided in the Appendix A. From Figure 15 it is observed that there are several pressure waves traveling along the feedback channels. At this instant, particularly high intensity pressure waves are seen at the upper feedback channel, lower-intensity waves are observed in the lower FC. It is important to realize that pressure waves traveling along a given FC have different strengths, see the different color intensities. Notice as well that the waves are observed at regular distances, the distance between two consecutive waves (d) should coincide with the sound speed divided by a particularly high amplitude oscillating or fluctuating frequency, $d=C/f=\sqrt{\gamma RT}/f$. Table 1 introduces, for the two power nozzle velocities and the four FC lengths studied, the maximum fluctuating frequency previously presented in Figure 13, for the case of 65 m/s and L9 the first harmonic of the frequency having the maximum amplitude is employed. Considering each particular frequency and the corresponding fluid temperature at the MC converging walls, the distance between two consecutive pressure waves is analytically determined. The distance separating two consecutive pressure waves is also measured and implemented in the same table. The agreement between the two distances is very good for all cases studied. At this point it can be stated that pressure waves traveling along the FCs are generated at the MC converging walls at a frequency coinciding with the jet fluctuating frequency for each case. As jet fluctuations sit on the top of the jet oscillations, pressure waves have a particular high intensity whenever both waves coincide (impinge) on the MC converging walls. Whenever the jet impinges perpendicular to the MC inclined wall, it helps in generating a higher-intensity pressure wave. For a power nozzle velocity of 97 m/s and a FC length 9L, Figure 16, it can be seen that the pressure waves appearing along the feedback channels are not generated regularly and their intensity differs temporally. As previously observed, fluid randomness is particularly high at long FCs and high speeds. These results fully coincide with the MC converging wall stagnation pressure presented in Figure 8d, where it is clearly seen that the degree of randomness is particularly high at 97 m/s. The randomness associated to the jet fluctuation can also be seen in Figure 11b for FC lengths of 3L and 9L.

The relation between the maximum and minimum pressure observed in Figure 15 shows a variation of about 6%, the fluid density changes about 7% while the maximum to minimum temperature variation is of about 2%. The maximum fluid velocity has increased by 40% versus the power nozzle one. For a power nozzle velocity of 97 m/s, the pressure variation observed in Figure 16 is of 15%, the density variation is about 16% and the temperature variation is about 4.5%. The percentage of velocity increase inside the MC is around 54% versus the power nozzle one. As expected, the percentage variation of the different parameters is higher for higher power nozzle velocities.

Another question which may arise is whether the maximum and minimum values of the velocity, pressure, density and temperature are affected by the feedback channel length. Despite the full information generated not being presented in this manuscript, it can be concluded that the maximum and minimum values of all these parameters remain very stable for all different feedback channel lengths studied, their variation was just found to depend on the power nozzle velocity.

## 6. Conclusions

The effect of the FC length in an angled FO geometry and under compressible subsonic flow conditions has been carefully studied. As feedback channel length increases, the outlet mass flow peak-to-peak amplitude and frequency decrease. The oscillation would eventually stop if a certain feedback channel length is overcome. See Figure 8 and Figure 9.

Pressure waves are generated when the main jet impinges alternatively at the mixing chamber converging walls. Under some conditions, pressure waves are also alternatively generated at the feedback channels outlet vertical internal walls. When this happens, the dynamic net-momentum, feedback channel mass flow and stagnation pressure signals are particularly scattered, this is due to the pressure waves appearing simultaneously on both sides of the feedback channels. See Figure 6b,c, Figure 7b,c and Figure 8.

At high feedback channel lengths, the typical jet oscillatory movement is almost gone, the jet mostly undergoes random fluctuations despite still performing low amplitude, low frequency oscillations. See Figure 8c,d. The amplitude of the fluctuations and their frequency increases with the feedback channel length increase, higher power nozzle inlet velocities also tend to increase the fluctuations amplitude and frequency. The jet has associated a high degree of randomness. See Figure 8 and Figure 9. The combination of the jet fluctuations and oscillations inside the MC generate pressure waves which travel along the feedback channels. Under these conditions, pressure waves are irregularly distributed, having as well different intensities. See Figure 6 and Figure 7 and particularly Figure 15 and Figure 16. Table 1 demonstrates that pressure waves are generated at very high frequencies, at fluctuating frequencies.

At small feedback channel lengths, the flow dynamics of all parameters are well-correlated, the stagnation pressure oscillations at the mixing chamber convergent walls drive the rest of the flow parameters, see Figure 8. At long feedback channel lengths and high power nozzle velocities, due to the highly chaotic fluctuations associated to the main jet inside the mixing chamber, the direct link between the stagnation pressure oscillations/fluctuations measured at the mixing chamber convergent walls and the dynamics of the rest of the parameters can still be directly established when performing the FFT of the different signals. Compare Figure 13 with Figure 10 and Figure 12.

The different high frequencies associated to the fluctuations appearing inside the MC and the FCs are not fully transferred to the flow at the FO outlets. It seems that the main jet is acting as a barrier preventing the displacement of these flow fluctuations to the external chamber. Compare Figure 10 which shows the outlet mass flow frequencies with Figure 12 and Figure 13. Clearly the high frequencies observed inside the MC and FCs are not seen at the outlet.

Under all conditions, the fluid suffers an expansion inside the mixing chamber. Vortical structures inside the MC and FCs are clearly observed, their sign changes alternatively with the jet oscillation inside the mixing chamber. Vortical structures can be observed in Figure 4, Figure 5, Figure 6 and Figure 7 and pressure fields in Figure 6 and Figure 7.

For the present fluidic oscillator configuration operating under compressible flow conditions, the oscillations are pressure-driven. The forces due to the mass flow flowing along the FCs and acting on the main jet at the feedback channels outlets, are an order of magnitude smaller than the ones due to the pressure. See Figure 14.

The authors strongly recommend to employ the methodology presented in this paper to determine in any feedback channel fluidic oscillator if oscillations are pressure- or mass flow-driven.

## Figures and Tables

**Figure 1 sensors-21-05768-f001:**
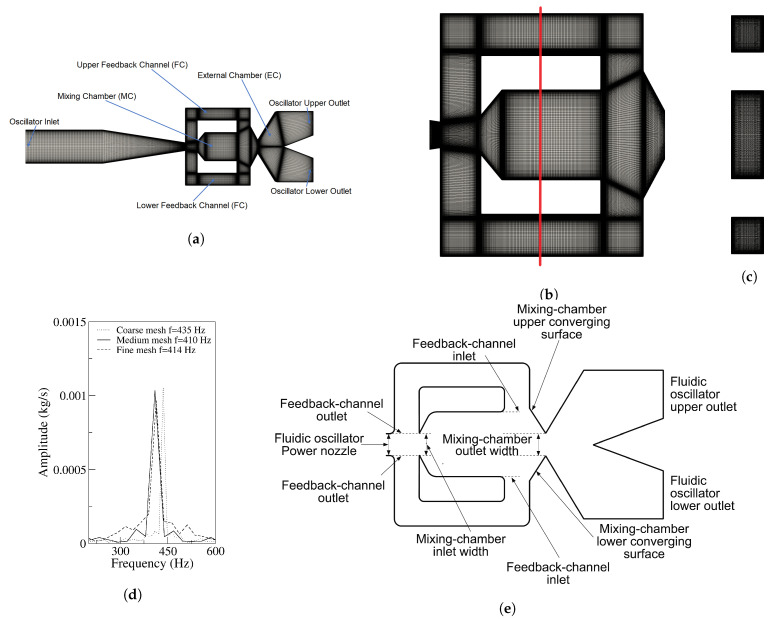
(**a**) Fluidic oscillator mesh, main view; (**b**) mixing chamber computational domain; (**c**) mesh side view; (**d**) fluidic oscillator outlet frequency for the three meshes studied; (**e**) mixing chamber main zones.

**Figure 2 sensors-21-05768-f002:**
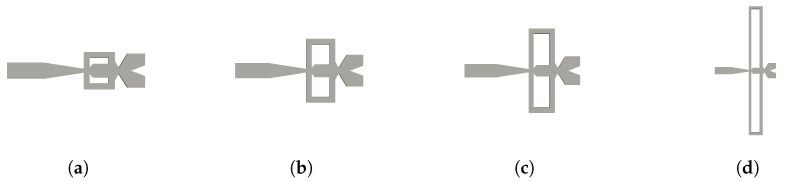
Baseline case fluidic oscillator with the four different feedback channel lengths studied. (**a**) Original feedback channel length L; (**b**) feedback channel length 2L; (**c**) feedback channel length 3L; (**d**) feedback channel length 9L.

**Figure 3 sensors-21-05768-f003:**
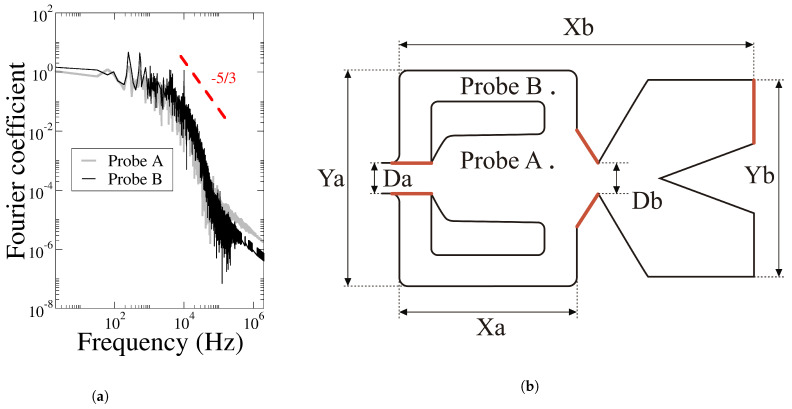
(**a**) Energy spectrum obtained from two probes located respectively at Probe A (*x* = 0.06642 m, *y* = 0.001693 m, *z* = 0.00195 m) and Probe B (*x* = 0.06642 m, *y* = 0.010157 m, *z* = 0.00195 m). (**b**) Probe locations and main dimensions.

**Figure 4 sensors-21-05768-f004:**
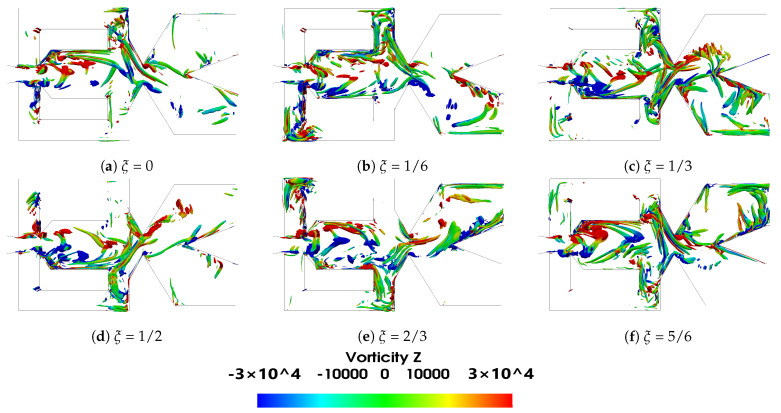
Q criterion representation at the mixing chamber period of oscillation divided into six equally spaced times, power nozzle velocity 65 m/s. *Q* = 5 × 10^8^. Baseline case.

**Figure 5 sensors-21-05768-f005:**
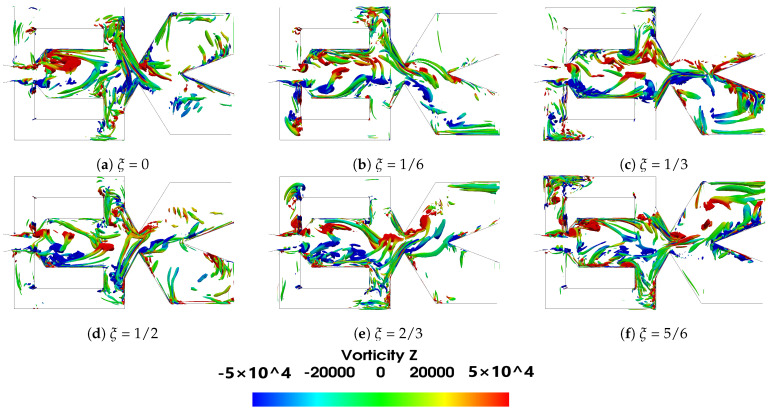
Q criterion representation at the mixing chamber period of oscillation divided into six equally spaced times, power nozzle velocity 97 m/s. *Q* = 11 × 10^8^. Baseline case.

**Figure 6 sensors-21-05768-f006:**
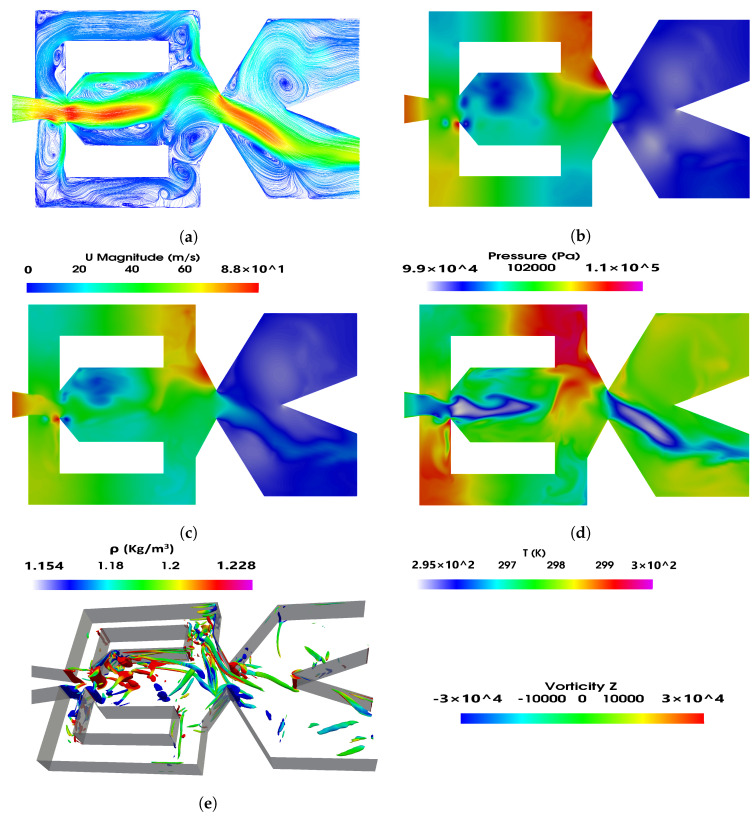
Mixing chamber instant velocity field (**a**), pressure distribution (**b**), density (**c**), temperature (**d**) and (**e**) Q criterion fields. Power nozzle velocity 65 m/s. Baseline case.

**Figure 7 sensors-21-05768-f007:**
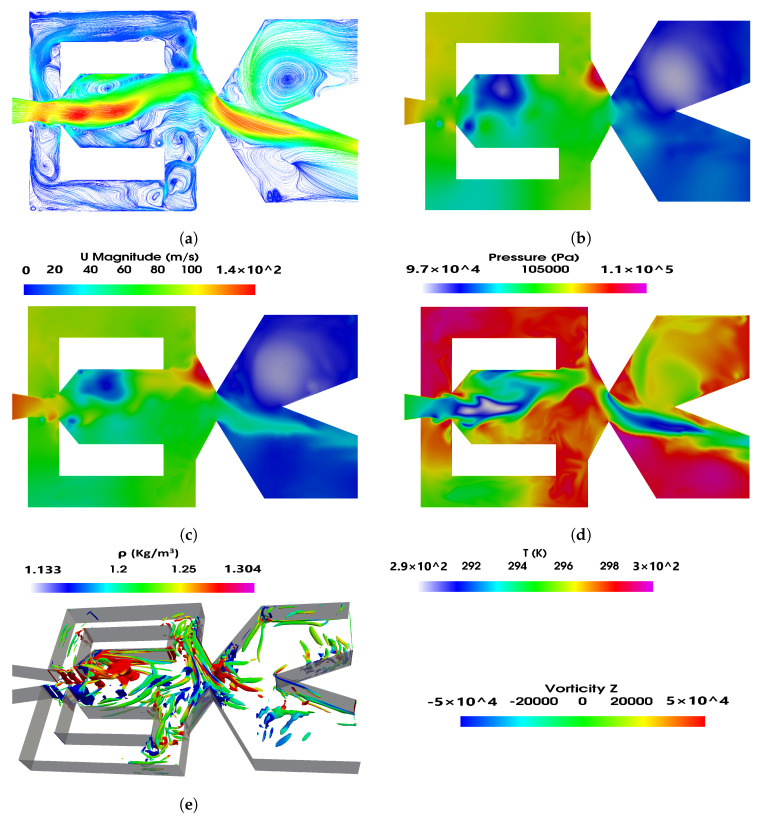
Mixing chamber instant velocity field (**a**), pressure distribution (**b**), density (**c**), temperature (**d**) and (**e**) Q criterion fields. Power nozzle velocity 97 m/s. Baseline case.

**Figure 8 sensors-21-05768-f008:**
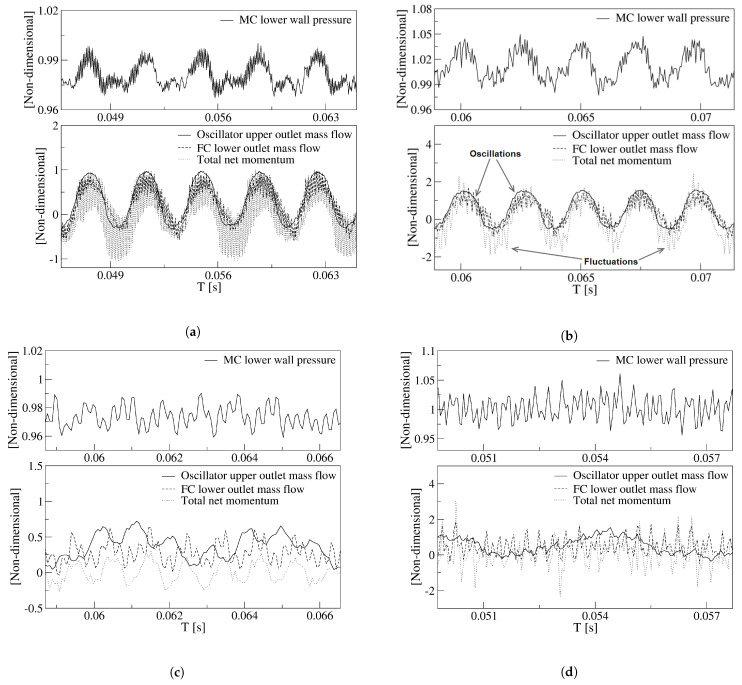
Non-dimensional pressure at the mixing chamber lower converging wall, oscillator mass flow, feedback channels mass flow and total net momentum applied to the incoming jet from both feedback channel outlets. (**a**,**b**) Baseline case. (**c**,**d**) Feedback channels length, 9L. (**a**,**c**) Power nozzle velocity, 65 m/s. (**b**,**d**) Power nozzle velocity, 97 m/s.

**Figure 9 sensors-21-05768-f009:**
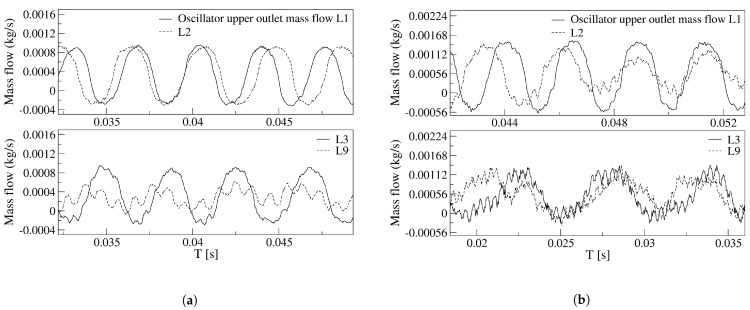
Fluidic oscillator upper outlet mass flow as a function of the different feedback channels length. (**a**) Power nozzle velocity, 65 m/s. (**b**) Power nozzle velocity, 97 m/s.

**Figure 10 sensors-21-05768-f010:**
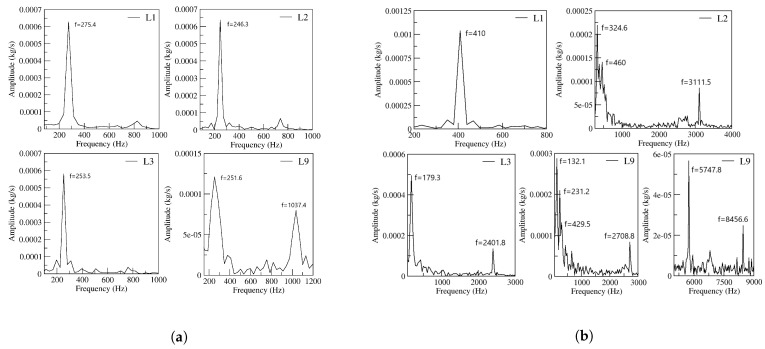
Fluidic Oscillator upper outlet mass flow frequency and peak-to-peak amplitude as a function of the power nozzle velocity and the feedback channel length. (**a**) Power nozzle velocity, 65 m/s. (**b**) Power nozzle velocity, 97 m/s.

**Figure 11 sensors-21-05768-f011:**
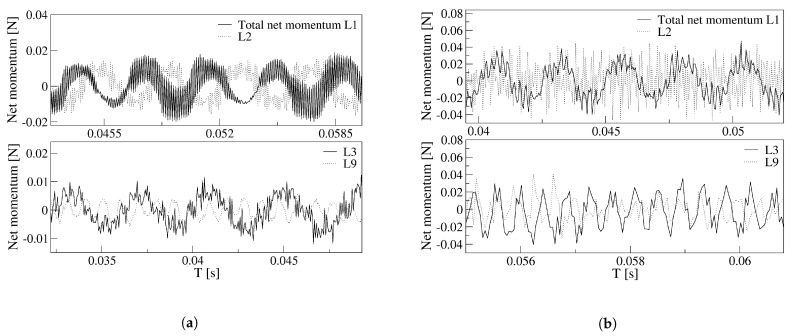
Net momentum applied to the jet entering the mixing chamber as a function of the different feedback channels length. (**a**) Power nozzle velocity, 65 m/s. (**b**) Power nozzle velocity, 97 m/s.

**Figure 12 sensors-21-05768-f012:**
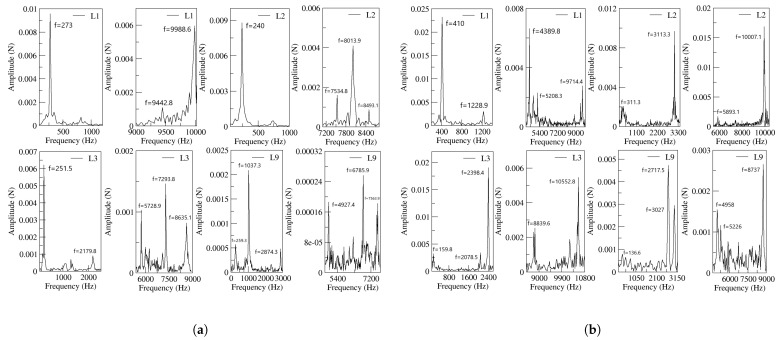
Frequencies obtained from the net-momentum signals measured at the feedback channels outlets, all feedback channels lengths are evaluated. (**a**) Power nozzle velocity, 65 m/s. (**b**) Power nozzle velocity, 97 m/s.

**Figure 13 sensors-21-05768-f013:**
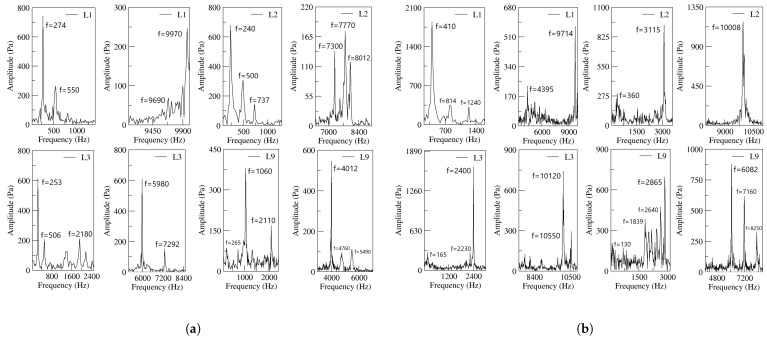
Frequencies obtained from the stagnation pressure measured at the MC inclined walls, all feedback channels lengths are evaluated. (**a**) Power nozzle velocity, 65 m/s. (**b**) Power nozzle velocity, 97 m/s.

**Figure 14 sensors-21-05768-f014:**
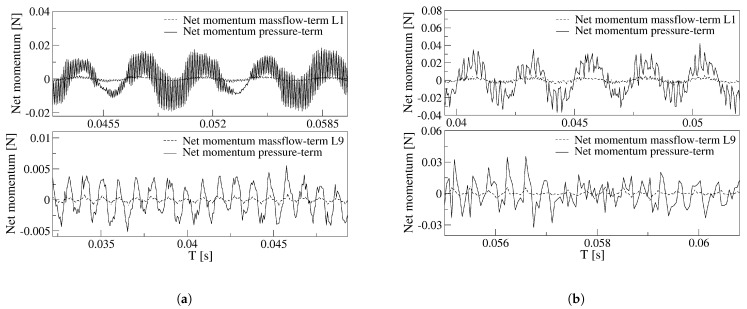
Comparison of the net momentum pressure term with the mass flow term, for the baseline case L1, and for the maximum (FC) length L9. (**a**) Power nozzle velocity, 65 m/s. (**b**) Power nozzle velocity, 97 m/s.

**Figure 15 sensors-21-05768-f015:**
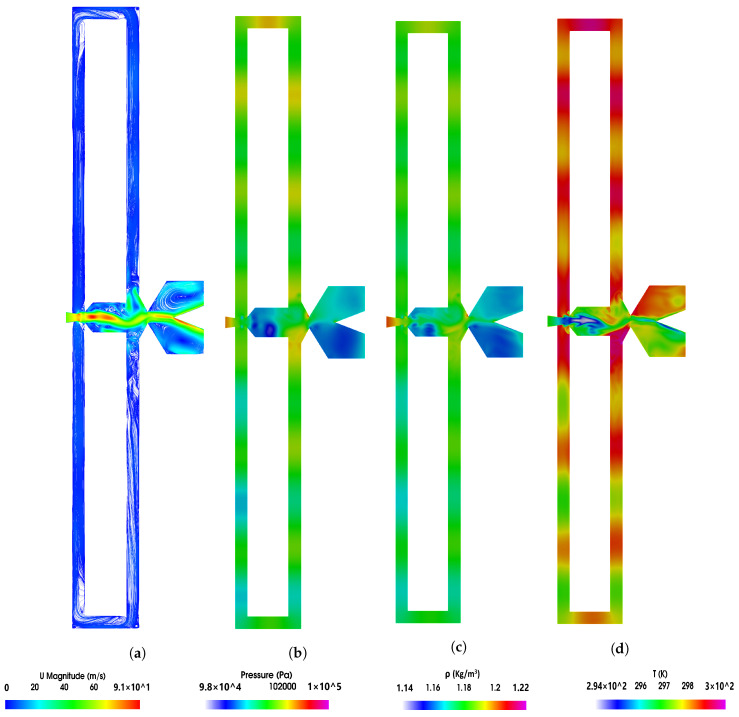
Feedback channels and mixing chamber instant velocity, pressure, density and temperature fields. Power nozzle velocity, 65 m/s, feedback channel length, L9.

**Figure 16 sensors-21-05768-f016:**
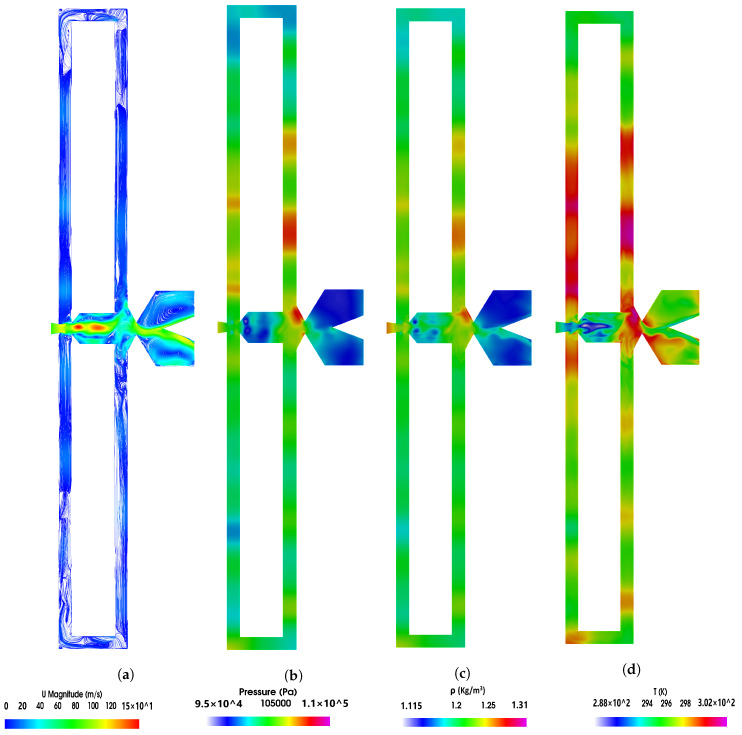
Feedback channels and mixing chamber instant velocity, pressure, density and temperature fields. Power nozzle velocity, 97 m/s, feedback channel length, L9.

**Table 1 sensors-21-05768-t001:** Comparison of the distance between two consecutive pressure waves measured versus the calculated distance.

	L1 (65 m/s)	L2 (65 m/s)	L3 (65 m/s)	L9 (65 m/s)	L1 (97 m/s)	L2 (97 m/s)	L3 (97 m/s)	L9 (97 m/s)
Max.frequency (Hz)	9970	8012	7292	8024	9714	10,008	10,550	8250
$d=C/f=\sqrt{\gamma RT}/f$ (mm)	34.6	43.2	47.4	43.2	35.7	34.7	32.9	42.1
*d*.measured (mm)	32	45	46	42	33	36	35	40

## Data Availability

Not applicable.

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
