# Peer review of "Fluidic Oscillators, Feedback Channel Effect under Compressible Flow Conditions"

_sensors, 2021, doi:10.3390/s21175768_

Round 1

Reviewer 1 Report

The study topic undertaken is timely, the purpose and scope of the study in the context of the literature review are clearly defined. The whole manuscript presents an appropriate standard of a scientific workshop for peer-reviewed publications. However, it finds some unclarities whose elimination will make it easier to interpret the results of the study.

1. 

The abstract needs to be modified because it does not contain details important from the point of view of a potential reader. Lacks numerical values to, for example, estimate the scale of the study. Main geometric parameters should be given. Reynolds numbers alone are not explicit. I find a similar situation in the conclusions. They contain correctly stated main observations but they are not supported by numerical values determined in chapter 5.

2. 

Referring to the comment regarding the abstract and the lack of dimensions of the test object, I think that the manuscript should include a simple summary of the main dimensions of the fluidic oscillators tested.

3. 

Is it possible to describe in a table or the text the parameters responsible for the visualization of streamlines shown in Fig. 6a, 7a, 15a, 16a? This would smooth the way for other researchers to replicate the study in the future.

4. 

A very interesting summary is shown in Figure 8. The question is raised what trend the analyzed parameters had for intermediate lengths of the feedback channel. For example, was there a visible increase in random fluctuations?

5. 

The location of the points where the analyzed parameters are determined should be more precisely described, e.g. as it was done in Figure 3, I consider this way as reference. In the manuscript, instead, one meets descriptions such as "the FFT of the stagnation pressure signal measured at the MC lower converging wall..." - line 467. This can be expressed more precisely.

6. 

Despite the great care taken in editing the manuscript, a few minor stumbling blocks were found, e.g. duplication of words or the way the unit of velocity was written (m / s instead of m/s). Please proofread the manuscript for these.

Author Response

Please see the file attached.

Thank you.

Reviewer 2 Report

The topic of this manuscript is interesting. The manuscript set up the three dimensional numerical simulations of the fluidic oscillators under compressible flow conditions, in the paper, four different feedback channel lengths and two inlet fluid velocities are considered. The paper may be value for the design and optimization of the fluidic oscillators, though there is no any improvement in theory and calculation techniques. The layout of the paper is logical and readable. With appropriate improvements, I agree to recommend it for publication in the journal.

Some suggestions for revision

  • The abstract section is too long. It is recommended to simplify it appropriately.
  • The flow velocity in FIG. 4 and FIG. 5 lacks units and is recommended to add.

Author Response

Please see the file attached.

Thank you

Reviewer 3 Report

The problem raised by the authors of the article entitled “Fluidic oscillators, feedback channel effect under compressible flow conditions”

The article focuses on the fluidic oscillators main performance, a problem which is not yet clarified is the understanding of the feedback channel effect on the oscillator outlet mass flow frequency and amplitude, specially under compressible flow conditions. For explaining that phenomena the Authors made three dimensional Direct Numerical Simulations under compressible flow conditions, are introduced in the present paper, four different feedback channel lengths and two inlet fluid velocities are considered. The results of their calculation are shown and discussed in this paper.

For a better understanding of the article, you should make some minor corrections, which are marked in attached file.

Please check whole text of paper especially equation or caption of the picture where multiplication sign was used *, please correct that sign on multiplication sign ×,

Please check the number on graph from 8 to 14, because for separation of number is used “,” please correct on dot “.”

The legend presented on maps form numerical simulation use wrong form of numbers, for example fig. 4 is -3.0e+4 should be -3×10^4, please check fig. 4, 5, 6, 7, 15 and 16.

Summarizing, the article requires a small correction. It should be published. I’m  recommend for publication after small remarks.

Author Response

Please see the file attached

Thank you.
